# CONTRASTIVE LEARNING OF STRUCTURED WORLD MODELS

**Thomas Kipf**
University of Amsterdam
t.n.kipf@uva.nl

**Elise van der Pol**
University of Amsterdam
UvA-Bosch Delta Lab
e.e.vanderpol@uva.nl

**Max Welling**
University of Amsterdam
CIFAR
m.welling@uva.nl

## ABSTRACT

A structured understanding of our world in terms of objects, relations, and hierarchies is an important component of human cognition. Learning such a structured world model from raw sensory data remains a challenge. As a step towards this goal, we introduce Contrastively-trained Structured World Models (C-SWMs). C-SWMs utilize a contrastive approach for representation learning in environments with compositional structure. We structure each state embedding as a set of object representations and their relations, modeled by a graph neural network. This allows objects to be discovered from raw pixel observations without direct supervision as part of the learning process. We evaluate C-SWMs on compositional environments involving multiple interacting objects that can be manipulated independently by an agent, simple Atari games, and a multi-object physics simulation. Our experiments demonstrate that C-SWMs can overcome limitations of models based on pixel reconstruction and outperform typical representatives of this model class in highly structured environments, while learning interpretable object-based representations.

## 1 INTRODUCTION

Compositional reasoning in terms of objects, relations, and actions is a central ability in human cognition (Spelke & Kinzler, 2007). This ability serves as a core motivation behind a range of recent works that aim at enriching machine learning models with the ability to disentangle scenes into objects, their properties, and relations between them (Chang et al., 2016; Battaglia et al., 2016; Watters et al., 2017; van Steenkiste et al., 2018; Kipf et al., 2018; Sun et al., 2018; 2019b; Xu et al., 2019). These structured neural models greatly facilitate predicting physical dynamics and the consequences of actions, and provide a strong inductive bias for generalization to novel environment situations, allowing models to answer counterfactual questions such as *"What would happen if I pushed this block instead of pulling it?"*.

Arriving at a structured description of the world in terms of objects and relations in the first place, however, is a challenging problem. While most methods in this area require some form of human annotation for the extraction of objects or relations, several recent works study the problem of object discovery from visual data in a completely unsupervised or self-supervised manner (Eslami et al., 2016; Greff et al., 2017; Nash et al., 2017; van Steenkiste et al., 2018; Kosiorek et al., 2018; Janner et al., 2019; Xu et al., 2019; Burgess et al., 2019; Greff et al., 2019; Engelcke et al., 2019). These methods follow a *generative* approach, i.e., they learn to discover object-based representations by performing visual predictions or reconstruction and by optimizing an objective in pixel space. Placing a loss in pixel space requires carefully trading off structural constraints on latent variables vs. accuracy of pixel-based reconstruction. Typical failure modes include ignoring visually small, but relevant features for predicting the future, such as a bullet in an Atari game (Kaiser et al., 2019), or wasting model capacity on visually rich, but otherwise potentially irrelevant features, such as static backgrounds.

To avoid such failure modes, we propose to adopt a *discriminative* approach using contrastive learning, which scores real against fake experiences in the form of state-action-state triples from an

experience buffer (Lin, 1992), in a similar fashion as typical graph embedding approaches score true facts in the form of entity-relation-entity triples against corrupted triples or fake facts.

We introduce Contrastively-trained Structured World Models (C-SWMs), a class of models for learning abstract state representations from observations in an environment. C-SWMs learn a set of abstract state variables, one for each object in a particular observation. Environment transitions are modeled using a graph neural network (Scarselli et al., 2009; Li et al., 2015; Kipf & Welling, 2016; Gilmer et al., 2017; Battaglia et al., 2018) that operates on latent abstract representations.

This paper further introduces a novel object-level contrastive loss for unsupervised learning of object-based representations. We arrive at this formulation by adapting methods for learning translational graph embeddings (Bordes et al., 2013; Wang et al., 2014) to our use case. By establishing a connection between contrastive learning of state abstractions (François-Lavet et al., 2018; Thomas et al., 2018) and relational graph embeddings (Nickel et al., 2016a), we hope to provide inspiration and guidance for future model improvements in both fields.

In a set of experiments, where we use a novel ranking-based evaluation strategy, we demonstrate that C-SWMs learn interpretable object-level state abstractions, accurately learn to predict state transitions many steps into the future, demonstrate combinatorial generalization to novel environment configurations and learn to identify objects from scenes without supervision.

## 2  STRUCTURED WORLD MODELS

Our goal is to learn an object-oriented abstraction of a particular observation or environment state. In addition, we would like to learn an action-conditioned transition model of the environment that takes object representations and their relations and interactions into account.

We start by introducing the general framework for contrastive learning of state abstractions and transition models *without* object factorization in Sections 2.1–2.2, and in the following describe a variant that utilizes object-factorized state representations, which we term a Structured World Model.

### 2.1  STATE ABSTRACTION

We consider an *off-policy* setting, where we operate solely on a buffer of offline experience, e.g., obtained from an exploration policy. Formally, this *experience buffer* $\mathcal{B} = \{(s_t, a_t, s_{t+1})\}_{t=1}^{T}$ contains $T$ tuples of states $s_t \in \mathcal{S}$, actions $a_t \in \mathcal{A}$, and follow-up states $s_{t+1} \in \mathcal{S}$, which are reached after taking action $a_t$. We do not consider rewards as part of our framework for simplicity.

Our goal is to learn abstract or *latent* representations $z_t \in \mathcal{Z}$ of environment states $s_t \in \mathcal{S}$ that discard any information which is not necessary to predict the abstract representation of the follow-up state $z_{t+1} \in \mathcal{Z}$ after taking action $a_t$. Formally, we have an *encoder* $E : \mathcal{S} \rightarrow \mathcal{Z}$ which maps observed states to abstract state representations and a *transition model* $T : \mathcal{Z} \times \mathcal{A} \rightarrow \mathcal{Z}$ operating solely on abstract state representations.

### 2.2  CONTRASTIVE LEARNING

Our starting point is the graph embedding method TransE (Bordes et al., 2013): TransE embeds facts from a knowledge base $\mathcal{K} = \{(e_t, r_t, o_t)\}_{t=1}^{T}$, which consists of entity-relation-entity triples $(e_t, r_t, o_t)$, where $e_t$ is the subject entity (analogous to the source state $s_t$ in our case), $r_t$ is the relation (analogous to the action $a_t$ in our experience buffer), and $o_t$ is the object entity (analogous to the target state $s_{t+1}$).

TransE defines the energy of a triple $(e_t, r_t, o_t)$ as $H = d(F(e_t) + G(r_t), F(o_t))$, where $F$ (and $G$) are embedding functions that map discrete entities (and relations) to $\mathbb{R}^D$, where $D$ is the dimensionality of the embedding space, and $d(\cdot, \cdot)$ denotes the squared Euclidean distance. Training is carried out with an energy-based hinge loss (LeCun et al., 2006), with negative samples obtained by replacing the entities in a fact with random entities from the knowledge base.

We can port TransE to our setting with only minor modifications. As the *effect* of an action is in general not independent of the source state, we replace $G(r_t)$ with $T(z_t, a_t)$, i.e., with the transition

function, conditioned on both the action and the (embedded) source state via $z_t = E(s_t)$. The overall energy of a state-action-state triple then can be defined as follows: $H = d(z_t + T(z_t, a_t), z_{t+1})$.

This additive form of the transition model provides a strong inductive bias for modeling effects of actions in the environment as translations in the abstract state space. Alternatively, one could model effects as linear transformations or rotations in the abstract state space, which motivates the use of a graph embedding method such as RESCAL (Nickel et al., 2011), CompleX (Trouillon et al., 2016), or HolE (Nickel et al., 2016b).

With the aforementioned modifications, we arrive at the following energy-based hinge loss:

$$\mathcal{L} = d(z_t + T(z_t, a_t), z_{t+1}) + \max(0, \gamma - d(\tilde{z}_t, z_{t+1})), \tag{1}$$

defined for a single $(s_t, a_t, s_{t+1})$ with a corrupted abstract state $\tilde{z}_t = E(\tilde{s}_t)$. $\tilde{s}_t$ is sampled at random from the experience buffer. The margin $\gamma$ is a hyperparameter for which we found $\gamma = 1$ to be a good choice. Unlike Bordes et al. (2013), we place the hinge only on the negative term instead of on the full loss and we do not constrain the norm of the abstract states $z_t$, which we found to work better in our context (see Appendix A.3). The overall loss is to be understood as an expectation of the above over samples from the experience buffer $\mathcal{B}$.

## 2.3 OBJECT-ORIENTED STATE FACTORIZATION

Our goal is to take into account the compositional nature of visual scenes, and hence we would like to learn a relational and object-oriented model of the environment that operates on a factored abstract state space $\mathcal{Z} = \mathcal{Z}_1 \times \ldots \times \mathcal{Z}_K$, where $K$ is the number of available object slots. We further assume an object-factorized action space $\mathcal{A} = \mathcal{A}_1 \times \ldots \times \mathcal{A}_K$. This factorization ensures that each object is independently represented and it allows for efficient sharing of model parameters across objects in the transition model. This serves as a strong inductive bias for better generalization to novel scenes and facilitates learning and object discovery. The overall C-SWM model architecture using object-factorized representations is shown in Figure 1.

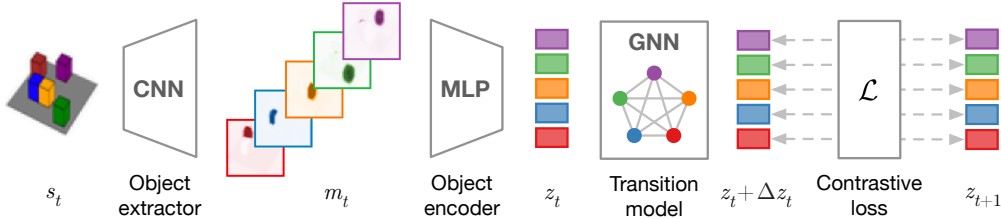

Figure 1: The C-SWM model is composed of the following components: 1) a CNN-based object extractor, 2) an MLP-based object encoder, 3) a GNN-based relational transition model, and 4) an object-factorized contrastive loss. Colored blocks denote abstract states for a particular object.

**Encoder and Object Extractor** We split the encoder into two separate modules: 1) a CNN-based object extractor $E_{\text{ext}}$, and 2) an MLP-based object encoder $E_{\text{enc}}$. The object extractor module is a CNN operating directly on image-based observations from the environment with $K$ feature maps in its last layer. Each feature map $m_t^k = [E_{\text{ext}}(s_t)]_k$ can be interpreted as an object mask corresponding to one particular object slot, where $[\ldots]_k$ denotes selection of the $k$-th feature map. For simplicity, we only assign a single feature map per object slot which sufficed for the experiments considered in this work (see Appendix A.4). To allow for encoding of more complex object features (other than, e.g., position/velocity), the object extractor can be adapted to produce multiple feature maps per object slot. After the object extractor module, we flatten each feature map $m_t^k$ (object mask) and feed it into the object encoder $E_{\text{enc}}$. The object encoder shares weights across objects and returns an abstract state representation: $z_t^k = E_{\text{enc}}(m_t^k)$ with $z_t^k \in \mathcal{Z}_k$. We set $\mathcal{Z}_k = \mathbb{R}^D$ in the following, where $D$ is a hyperparameter.

**Relational Transition Model** We implement the transition model as a graph neural network (Scarselli et al., 2009; Li et al., 2015; Kipf & Welling, 2016; Battaglia et al., 2016; Gilmer et al., 2017; Battaglia et al., 2018), which allows us to model pairwise interactions between object states

while being invariant to the order in which objects are represented. After the encoder stage, we have an abstract state description $z_t^k \in \mathcal{Z}_k$ and an action $a_t^k \in \mathcal{A}_k$ for every object in the scene. We represent actions as one-hot vectors (or a vector of zeros if no action is applied to a particular object), but note that other choices are possible, e.g., for continuous action spaces. The transition function then takes as input the tuple of object representations $z_t = (z_t^1, \ldots, z_t^K)$ and actions $a_t = (a_t^1, \ldots, a_t^K)$ at a particular time step:

$$\Delta z_t = T(z_t, a_t) = \text{GNN}(\{(z_t^k, a_t^k)\}_{k=1}^K). \tag{2}$$

$T(z_t, a_t)$ is implemented as a graph neural network (GNN) that takes $z_t^k$ as input node features. The model predicts updates $\Delta z_t = (\Delta z_t^1, \ldots, \Delta z_t^K)$. The object representations for the next time step are obtained via $z_{t+1} = (z_t^1 + \Delta z_t^1, \ldots, z_t^K + \Delta z_t^K)$. The GNN consists of node update functions $f_{\text{node}}$ and edge update functions $f_{\text{edge}}$ with shared parameters across all nodes and edges. These functions are implemented as MLPs and we choose the following form of message passing updates:

$$e_t^{(i,j)} = f_{\text{edge}}([z_t^i, z_t^j]) \tag{3}$$

$$\Delta z_t^j = f_{\text{node}}([z_t^j, a_t^j, \sum_{i \neq j} e_t^{(i,j)}]), \tag{4}$$

where $e_t^{(i,j)}$ is an intermediate representation of the edge or interaction between nodes $i$ and $j$. This corresponds to a single round of node-to-edge and edge-to-node message passing. Alternatively, one could apply multiple rounds of message passing, but we did not find this to be necessary for the experiments considered in this work. Note that this update rule corresponds to message passing on a fully-connected scene graph, which is $\mathcal{O}(K^2)$. This can be reduced to linear complexity by reducing connectivity to nearest neighbors in the abstract state space, which we leave for future work. We denote the output of the transition function for the $k$-th object as $\Delta z_t^k = T^k(z_t, a_t)$ in the following.

**Multi-object Contrastive Loss**     We only need to change the energy function to take the factorization of the abstract state space into account, which yields the following energy $H$ for positive triples and $\tilde{H}$ for negative samples:

$$H = \frac{1}{K}\sum_{k=1}^K d(z_t^k + T^k(z_t, a_t), z_{t+1}^k), \quad \tilde{H} = \frac{1}{K}\sum_{k=1}^K d(\tilde{z}_t^k, z_{t+1}^k), \tag{5}$$

where $\tilde{z}_t^k$ is the $k$-th object representation of the negative state sample $\tilde{z}_t = E(\tilde{s}_t)$. The overall contrastive loss for a single state-action-state sample from the experience buffer then takes the form:

$$\mathcal{L} = H + \max(0, \gamma - \tilde{H}). \tag{6}$$

## 3  RELATED WORK

For coverage of related work in the area of object discovery with autoencoder-based models, we refer the reader to the Introduction section. We further discuss related work on relational graph embeddings in Section 2.2.

**Structured Models of Environments**     Recent work on modeling structured environments such as interacting multi-object or multi-agent systems has made great strides in improving predictive accuracy by explicitly taking into account the structured nature of such systems (Sukhbaatar et al., 2016; Chang et al., 2016; Battaglia et al., 2016; Watters et al., 2017; Hoshen, 2017; Wang et al., 2018; van Steenkiste et al., 2018; Kipf et al., 2018; Sanchez-Gonzalez et al., 2018; Xu et al., 2019). These methods generally make use of some form of graph neural network, where node update functions model the dynamics of individual objects, parts or agents and edge update functions model their interactions and relations. Several recent works succeed in learning such structured models directly from pixels (Watters et al., 2017; van Steenkiste et al., 2018; Xu et al., 2019; Watters et al., 2019), but in contrast to our work rely on pixel-based loss functions. The latest example in this line of research is the COBRA model (Watters et al., 2019), which learns an action-conditioned transition policy on object representations obtained from an unsupervised object discovery model (Burgess et al., 2019). Unlike C-SWM, COBRA does not model interactions between object slots and relies on a pixel-based loss for training. Our object encoder, however, is more limited and utilizing an iterative object encoding process such as in MONet (Burgess et al., 2019) would be interesting for future work.

**Contrastive Learning** Contrastive learning methods are widely used in the field of graph representation learning (Bordes et al., 2013; Perozzi et al., 2014; Grover & Leskovec, 2016; Bordes et al., 2013; Schlichtkrull et al., 2018; Veličković et al., 2018), and for learning word representations (Mnih & Teh, 2012; Mikolov et al., 2013). The main idea is to construct pairs of related data examples (positive examples, e.g., connected by an edge in a graph or co-occuring words in a sentence) and pairs of unrelated or corrupted data examples (negative examples), and use a loss function that scores positive and negative pairs in a different way. Most energy-based losses (LeCun et al., 2006) are suitable for this task. Recent works (Oord et al., 2018; Hjelm et al., 2018; Hénaff et al., 2019; Sun et al., 2019a; Anand et al., 2019) connect objectives of this kind to the principle of learning representations by maximizing mutual information between data and learned representations, and successfully apply these methods to image, speech, and video data.

**State Representation Learning** State representation learning in environments similar to ours is often approached by models based on autoencoders (Corneil et al., 2018; Watter et al., 2015; Ha & Schmidhuber, 2018; Hafner et al., 2019; Laversanne-Finot et al., 2018) or via adversarial learning (Kurutach et al., 2018; Wang et al., 2019). Some recent methods learn state representations without requiring a decoder back into pixel space. Examples include the selectivity objective in Thomas et al. (2018), the contrastive objective in François-Lavet et al. (2018), the mutual information objective in Anand et al. (2019), the distribution matching objective in Gelada et al. (2019) or using causality-based losses and physical priors in latent space (Jonschkowski & Brock, 2015; Ehrhardt et al., 2018). Most notably, Ehrhardt et al. (2018) propose a method to learn an object detection module and a physics module jointly from raw video data without pixel-based losses. This approach, however, can only track a single object at a time and requires careful balancing of multiple loss functions.

## 4 EXPERIMENTS

Our goal of this experimental section is to verify whether C-SWMs can 1) learn to discover object representations from environment interactions without supervision, 2) learn an accurate transition model in latent space, and 3) generalize to novel, unseen scenes. Our implementation is available under https://github.com/tkipf/c-swm.

### 4.1 ENVIRONMENTS

We evaluate C-SWMs on two novel grid world environments (2D shapes and 3D blocks) involving multiple interacting objects that can be manipulated independently by an agent, two Atari 2600 games (Atari Pong and Space Invaders), and a multi-object physics simulation (3-body physics). See Figure 2 for example observations.

For all environments, we use a random policy to collect experience for both training and evaluation. Observations are provided as $50 \times 50 \times 3$ color images for the grid world environments and as $50 \times 50 \times 6$ tensors (two concatenated consecutive frames) for the Atari and 3-body physics environments. Additional details on environments and dataset creation can be found in Appendix B.

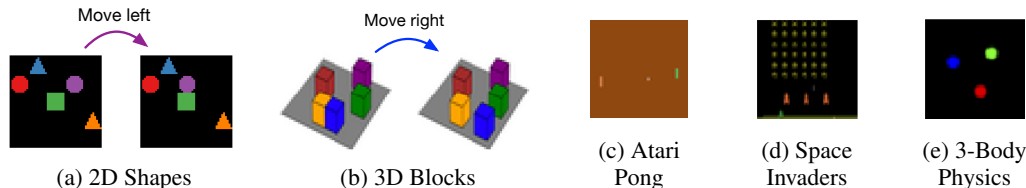

(a) 2D Shapes      (b) 3D Blocks      (c) Atari Pong      (d) Space Invaders      (e) 3-Body Physics

Figure 2: Example observations from block pushing environments (a–b), Atari 2600 games (c–d) and a 3-body gravitational physics simulation (e). In the grid worlds (a–b), each block can be independently moved into the four cardinal directions unless the target position is occupied by another block or outside of the scene. Best viewed in color.

## 4.2 Evaluation Metrics

In order to evaluate model performance directly in latent space, we make use of ranking metrics, which are commonly used for the evaluation of link prediction models, as in, e.g., Bordes et al. (2013). This allows us to assess the quality of learned representations directly without relying on auxiliary metrics such as pixel-based reconstruction losses, or performance in downstream tasks such as planning.

Given an observation encoded by the model and an action, we use the model to predict the representation of the next state, reached after taking the action in the environment. This predicted state representation is then compared to the encoded true observation after taking the action in the environment and a set of reference states (observations encoded by the model) obtained from the experience buffer. We measure and report both Hits at Rank 1 (H@1) and Mean Reciprocal Rank (MRR). Additional details on these evaluation metrics can be found in Appendix C.

## 4.3 Baselines

**Autoencoder-based World Models**   The predominant method for state representation learning is based on autoencoders, and often on the VAE (Kingma & Welling, 2013; Rezende et al., 2014) model in particular. This World Model baseline is inspired by Ha & Schmidhuber (2018) and uses either a deterministic autoencoder (AE) or a VAE to learn state representations. Finally, an MLP is used to predict the next state after taking an action.

**Physics As Inverse Graphics (PAIG)**   This model by Jaques et al. (2019) is based on an encoder-decoder architecture and trained with pixel-based reconstruction losses, but uses a differentiable physics engine in the latent space that operates on explicit position and velocity representations for each object. Thus, this model is only applicable to the 3-body physics environment.

## 4.4 Training and Evaluation Setting

We train C-SWMs on an experience buffer obtained by running a random policy on the respective environment. We choose 1000 episodes with 100 environment steps each for the grid world environments, 1000 episodes with 10 steps each for the Atari environments and 5000 episodes with 10 steps each for the 3-body physics environment.

For evaluation, we populate a separate experience buffer with 10 environment steps per episode and a total of 10.000 episodes for the grid world environments, 100 episodes for the Atari environments and 1000 episodes for the physics environment. For the Atari environments, we minimize train/test overlap by 'warm-starting' experience collection in these environments with random actions before we start populating the experience buffer (see Appendix B), and we ensure that not a single full test set episode coincides exactly with an episode from the training set. The state spaces of the grid world environments are large (approx. 6.4M unique states) and hence train/test coincidence of a full 10-step episode is unlikely. Overlap is similarly unlikely for the physics environment which has a continuous state space. Hence, performing well on these tasks will require some form of generalization to new environment configurations or an unseen sequence of states and actions.

All models are trained for 100 epochs (200 for Atari games) using the Adam (Kingma & Ba, 2014) optimizer with a learning rate of $5 \cdot 10^{-4}$ and a batch size of 1024 (512 for baselines with decoders due to higher memory demands, and 100 for PAIG as suggested by the authors). Model architecture details are provided in Appendix D.

## 4.5 Qualitative Results

We present qualitative results for the grid world environments in Figure 3 and for the 3-body physics environment in Figure 4. All results are obtained on hold-out test data. Further qualitative results (incl. on Atari games) can be found in Appendix A.

In the grid world environments, we can observe that C-SWM reliably discovers object-specific filters for a particular scene, without direct supervision. Further, each object is represented by two coordinates which correspond (up to a random linear transformation) to the true object position in

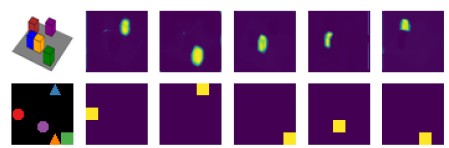

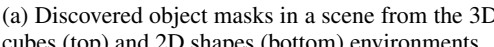

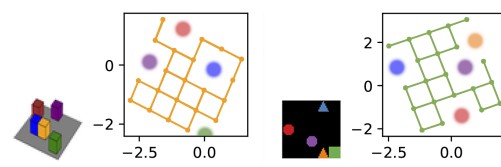

(a) Discovered object masks in a scene from the 3D cubes (top) and 2D shapes (bottom) environments.

(b) Learned abstract state transition graph of the yellow cube (left) and the green square (right), while keeping all other object positions fixed at test time.

Figure 3: Discovered object masks (left) and direct visualization of the 2D abstract state spaces and transition graphs for a single object (right) in the block pushing environments. Nodes denote state embeddings obtained from a test set experience buffer with random actions and edges are predicted transitions. The learned abstract state graph clearly captures the underlying grid structure of the environment both in terms of object-specific latent states and in terms of predicted transitions, but is randomly rotated and/or mirrored. The model further correctly captures that certain actions do not have an effect if a neighboring position is blocked by another object (shown as colored spheres), even though the transition model does not have access to visual inputs.

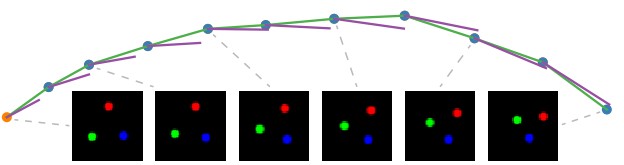

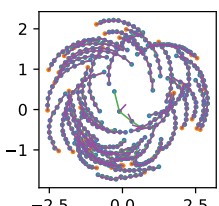

(a) Observations from 3-body gravitational physics simulation (bottom) and learned abstract state transition graph for a single object slot (top).

(b) Abstract state transition graph from 50 test episodes for single object slot.

Figure 4: Qualitative results for 3-body physics environment for a single representative test set episode (left) and for a dataset of 50 test episodes (right). The model learns to smoothly embed object trajectories, with the circular motion represented in the latent space (projected from four to two dimensions via PCA). In the abstract state transition graph, orange nodes denote starting states for a particular episode, green links correspond to ground truth transitions and violet links correspond to transitions predicted by the model. One trajectory (in the center) strongly deviates from typical trajectories seen during training, and the model struggles to predict the correct transition.

the scene. Although we choose a two-dimensional latent representation per object for easier visualization, we find that results remain unchanged if we increase the dimensionality of the latent representation. The edges in this learned abstract transition graph correspond to the effect of a particular action applied to the object. The structure of the learned latent representation accurately captures the underlying grid structure of the environment. We further find that the transition model, which only has access to latent representations, correctly captures whether an action has an effect or not, e.g., if a neighboring position is blocked by another object.

Similarly, we find that the model can learn object-specific encoders in the 3-body physics environment and can learn object-specific latent representations that track location and velocity of a particular object, while learning an accurate latent transition model that generalizes well to unseen environment instances.

## 4.6 QUANTITATIVE RESULTS

We set up quantitative experiments for evaluating the quality of both object discovery and the quality of the learned transition model. We compare against autoencoder baselines and model variants that do not represent the environment in an object-factorized manner, do not use a GNN, or do not make use of contrastive learning. Performing well under this evaluation setting requires some degree of (combinatorial) generalization to unseen environment instances.

We report ranking scores (in %) in latent space, after encoding source and target observations, and taking steps in the latent space using the learned model. Reported results are mean and standard error of scores over 4 runs on hold-out environment instances. Results are summarized in Table 1.

Table 1: Ranking results for multi-step prediction in latent space. Highest (mean) scores in **bold**.

| | Model | 1 Step | | 5 Steps | | 10 Steps | |
|---|---|---|---|---|---|---|---|
| | | H@1 | MRR | H@1 | MRR | H@1 | MRR |
| 2D SHAPES | **C-SWM** | $\mathbf{100}_{\pm 0.0}$ | $\mathbf{100}_{\pm 0.0}$ | $\mathbf{100}_{\pm 0.0}$ | $\mathbf{100}_{\pm 0.0}$ | $\mathbf{99.9}_{\pm 0.0}$ | $\mathbf{100}_{\pm 0.0}$ |
| | – latent GNN | $99.9_{\pm 0.0}$ | $\mathbf{100}_{\pm 0.0}$ | $97.4_{\pm 0.1}$ | $98.4_{\pm 0.0}$ | $89.7_{\pm 0.3}$ | $93.1_{\pm 0.2}$ |
| | – factored states | $54.5_{\pm 18.1}$ | $65.0_{\pm 15.9}$ | $34.4_{\pm 16.0}$ | $47.4_{\pm 16.0}$ | $24.1_{\pm 11.2}$ | $37.0_{\pm 12.1}$ |
| | – contrastive loss | $49.9_{\pm 0.9}$ | $55.2_{\pm 0.9}$ | $6.5_{\pm 0.5}$ | $9.3_{\pm 0.7}$ | $1.4_{\pm 0.1}$ | $2.6_{\pm 0.2}$ |
| | World Model (AE) | $98.7_{\pm 0.5}$ | $99.2_{\pm 0.3}$ | $36.1_{\pm 8.1}$ | $44.1_{\pm 8.1}$ | $6.5_{\pm 2.6}$ | $10.5_{\pm 3.6}$ |
| | World Model (VAE) | $94.2_{\pm 1.0}$ | $96.4_{\pm 0.6}$ | $14.1_{\pm 1.1}$ | $21.4_{\pm 1.4}$ | $1.4_{\pm 0.2}$ | $3.5_{\pm 0.4}$ |
| 3D BLOCKS | **C-SWM** | $\mathbf{99.9}_{\pm 0.0}$ | $\mathbf{100}_{\pm 0.0}$ | $\mathbf{99.9}_{\pm 0.0}$ | $\mathbf{100}_{\pm 0.0}$ | $\mathbf{99.9}_{\pm 0.0}$ | $\mathbf{99.9}_{\pm 0.0}$ |
| | – latent GNN | $\mathbf{99.9}_{\pm 0.0}$ | $99.9_{\pm 0.0}$ | $96.3_{\pm 0.4}$ | $97.7_{\pm 0.3}$ | $86.0_{\pm 1.8}$ | $90.2_{\pm 1.5}$ |
| | – factored states | $74.2_{\pm 9.3}$ | $82.5_{\pm 8.3}$ | $48.7_{\pm 12.9}$ | $62.6_{\pm 13.0}$ | $65.8_{\pm 14.0}$ | $49.6_{\pm 11.0}$ |
| | – contrastive loss | $48.9_{\pm 16.8}$ | $52.5_{\pm 17.8}$ | $12.2_{\pm 5.8}$ | $16.3_{\pm 7.1}$ | $3.1_{\pm 1.9}$ | $5.3_{\pm 2.8}$ |
| | World Model (AE) | $93.5_{\pm 0.8}$ | $95.6_{\pm 0.6}$ | $26.7_{\pm 0.7}$ | $35.6_{\pm 0.8}$ | $4.0_{\pm 0.2}$ | $7.6_{\pm 0.3}$ |
| | World Model (VAE) | $90.9_{\pm 0.7}$ | $94.2_{\pm 0.6}$ | $31.3_{\pm 2.3}$ | $41.8_{\pm 2.3}$ | $7.2_{\pm 0.9}$ | $12.9_{\pm 1.3}$ |
| ATARI PONG | **C-SWM** ($K = 5$) | $20.5_{\pm 3.5}$ | $41.8_{\pm 2.9}$ | $9.5_{\pm 2.2}$ | $22.2_{\pm 3.3}$ | $5.3_{\pm 1.6}$ | $15.8_{\pm 2.8}$ |
| | **C-SWM** ($K = 3$) | $34.8_{\pm 5.3}$ | $54.3_{\pm 5.2}$ | $12.8_{\pm 3.4}$ | $28.1_{\pm 4.2}$ | $9.5_{\pm 1.7}$ | $21.1_{\pm 2.8}$ |
| | **C-SWM** ($K = 1$) | $\mathbf{36.5}_{\pm 5.6}$ | $\mathbf{56.2}_{\pm 6.2}$ | $\mathbf{18.3}_{\pm 1.9}$ | $\mathbf{35.7}_{\pm 2.3}$ | $\mathbf{11.5}_{\pm 1.0}$ | $\mathbf{26.0}_{\pm 1.2}$ |
| | World Model (AE) | $23.8_{\pm 3.3}$ | $44.7_{\pm 2.4}$ | $1.7_{\pm 0.5}$ | $8.0_{\pm 0.5}$ | $1.2_{\pm 0.8}$ | $5.3_{\pm 0.8}$ |
| | World Model (VAE) | $1.0_{\pm 0.0}$ | $5.1_{\pm 0.1}$ | $1.0_{\pm 0.0}$ | $5.2_{\pm 0.0}$ | $1.0_{\pm 0.0}$ | $5.2_{\pm 0.0}$ |
| SPACE INVADERS | **C-SWM** ($K = 5$) | $\mathbf{48.5}_{\pm 7.0}$ | $\mathbf{66.1}_{\pm 6.6}$ | $\mathbf{16.8}_{\pm 2.7}$ | $\mathbf{35.7}_{\pm 3.7}$ | $\mathbf{11.8}_{\pm 3.0}$ | $\mathbf{26.0}_{\pm 4.1}$ |
| | **C-SWM** ($K = 3$) | $46.2_{\pm 13.0}$ | $62.3_{\pm 11.5}$ | $10.8_{\pm 3.7}$ | $28.5_{\pm 5.8}$ | $6.0_{\pm 0.4}$ | $20.9_{\pm 0.9}$ |
| | **C-SWM** ($K = 1$) | $31.5_{\pm 13.1}$ | $48.6_{\pm 11.8}$ | $10.0_{\pm 2.3}$ | $23.9_{\pm 3.6}$ | $6.0_{\pm 1.7}$ | $19.8_{\pm 3.3}$ |
| | World Model (AE) | $40.2_{\pm 3.6}$ | $59.6_{\pm 3.5}$ | $5.2_{\pm 1.1}$ | $14.1_{\pm 2.0}$ | $3.8_{\pm 0.8}$ | $10.4_{\pm 1.3}$ |
| | World Model (VAE) | $1.0_{\pm 0.0}$ | $5.3_{\pm 0.1}$ | $0.8_{\pm 0.2}$ | $5.2_{\pm 0.0}$ | $1.0_{\pm 0.0}$ | $5.2_{\pm 0.0}$ |
| 3-BODY PHYSICS | **C-SWM** | $\mathbf{100}_{\pm 0.0}$ | $\mathbf{100}_{\pm 0.0}$ | $97.2_{\pm 0.9}$ | $98.5_{\pm 0.5}$ | $\mathbf{75.5}_{\pm 4.7}$ | $\mathbf{85.2}_{\pm 3.1}$ |
| | World Model (AE) | $\mathbf{100}_{\pm 0.0}$ | $\mathbf{100}_{\pm 0.0}$ | $\mathbf{97.7}_{\pm 0.3}$ | $\mathbf{98.8}_{\pm 0.2}$ | $67.9_{\pm 2.4}$ | $78.4_{\pm 1.8}$ |
| | World Model (VAE) | $\mathbf{100}_{\pm 0.0}$ | $\mathbf{100}_{\pm 0.0}$ | $83.1_{\pm 2.5}$ | $90.3_{\pm 1.6}$ | $23.6_{\pm 4.2}$ | $37.5_{\pm 4.8}$ |
| | Physics WM (PAIG) | $89.2_{\pm 3.5}$ | $90.7_{\pm 3.4}$ | $57.7_{\pm 12.0}$ | $63.1_{\pm 11.1}$ | $25.1_{\pm 13.0}$ | $33.1_{\pm 13.4}$ |

We find that baselines that make use of reconstruction losses in pixel space (incl. the C-SWM model variant without contrastive loss) typically generalize less well to unseen scenes and learn a latent space configuration that makes it difficult for the transition model to learn the correct transition function. See Appendix A.1 for a visual analysis of such latent state transition graphs. This effect appears to be even stronger when using a VAE-based World Model, where the prior puts further constraints on the latent representations. C-SWM recovers this structure well, see Figure 3.

On the grid-world environments (2D shapes and 3D blocks), C-SWM models latent transitions almost perfectly, which requires taking interactions between latent representations of objects into account. Removing the interaction component, i.e., replacing the latent GNN with an object-wise MLP, makes the model insensitive to pairwise interactions and hence the ability to predict future states deteriorates. Similarly, if we remove the state factorization, the model has difficulties generalizing to unseen environment configurations. We further explore variants of the grid-world environments in Appendix A.5, where 1) certain actions have no effect, and 2) one object moves randomly and independent of agent actions, to test the robustness of our approach.

For the Atari 2600 experiments, we find that results can have a high variance, and that the task is more difficult, as both the World Model baseline and C-SWM struggle to make perfect long-term predictions. While for Space Invaders, a large number of object slots ($K = 5$) appears to be beneficial, C-SWM achieves best results with only a single object slot in Atari Pong. This suggests that one should determine the optimal value of $K$ based on performance on a validation set if it is not known a-priori. Using an iterative object encoding mechanism, such as in MONet (Burgess et al., 2019), would enable the model to assign 'empty' slots which could improve robustness w.r.t. the choice of $K$, which we leave for future work.

We find that both C-SWMs and the autoencoder-based World Model baseline excel at short-term predictions in the 3-body physics environment, with C-SWM having a slight edge in the 10 step prediction setting. Under our evaluation setting, the PAIG baseline (Jaques et al., 2019) underperforms using the hyperparameter setting recommended by the authors. Note that we do not tune hyperparameters of C-SWM separately for this task and use the same settings as in other environments.

### 4.7 LIMITATIONS

**Instance Disambiguation**    In our experiments, we chose a simple feed-forward CNN architecture for the object extractor module. This type of architecture cannot disambiguate multiple instances of the same object present in one scene and relies on distinct visual features or labels (e.g., *the green square*) for object extraction. To better handle scenes which contain potentially multiple copies of the same object (e.g., in the Atari Space Invaders game), one would require some form of iterative disambiguation procedure to break symmetries and dynamically bind individual objects to slots or *object files* (Kahneman & Treisman, 1984; Kahneman et al., 1992), such as in the style of dynamic routing (Sabour et al., 2017), iterative inference (Greff et al., 2019; Engelcke et al., 2019) or sequential masking (Burgess et al., 2019; Kipf et al., 2019).

**Stochasticity & Markov Assumption**    Our formulation of C-SWMs does not take into account stochasticity in environment transitions or observations, and hence is limited to fully deterministic worlds. A probabilistic extension of C-SWMs is an interesting avenue for future work. For simplicity, we make the Markov assumption: state and action contain all the information necessary to predict the next state. This allows us to look at single state-action-state triples in isolation. To go beyond this limitation, one would require some form of memory mechanism, such as an RNN as part of the model architecture, which we leave for future work.

## 5    CONCLUSIONS

Structured world models offer compelling advantages over pure connectionist methods, by enabling stronger inductive biases for generalization, without necessarily constraining the generality of the model: for example, the contrastively trained model on the 3-body physics environment is free to store identical representations in each object slot and ignore pairwise interactions, i.e., an unstructured world model still exists as a special case. Experimentally, we find that C-SWMs make effective use of this additional structure, likely because it allows for a transition model of significantly lower complexity, and learn object-oriented models that generalize better to unseen situations.

We are excited about the prospect of using C-SWMs for model-based planning and reinforcement learning in future work, where object-oriented representations will likely allow for more accurate counterfactual reasoning about effects of actions and novel interactions in the environment. We further hope to inspire future work to think beyond autoencoder-based approaches for object-based, structured representation learning, and to address some of the limitations outlined in this paper.

### ACKNOWLEDGEMENTS

We would like to thank Marco Federici and Adam Kosiorek for helpful discussions. We would further like to thank the anonymous reviewers for valuable feedback. T.K. acknowledges funding by SAP SE.

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

## A    ADDITIONAL RESULTS AND DISCUSSION

### A.1    OBJECT-SPECIFIC REPRESENTATIONS

We visualize abstract state transition graphs separated by object slot for the 3D cubes environment in Figure 5. Discovered object representations in the 2D shapes dataset (not shown) are qualitatively very similar. We apply the same visualization technique to the model variant without contrastive loss, which is instead trained with a decoder model and a loss in pixel space. See Figure 6 for this baseline and note that the regular structure in the latent space is lost, which makes it difficult for the transition model to learn transitions which generalize to unseen environment instances.

Qualitative results for the 3-body physics dataset are summarized in Figures 7 and 8 for two different random seeds.

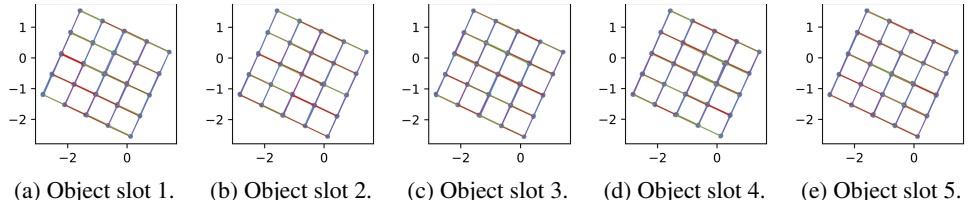

(a) Object slot 1.    (b) Object slot 2.    (c) Object slot 3.    (d) Object slot 4.    (e) Object slot 5.

Figure 5: Abstract state transition graphs per object slot for a trained C-SWM model on the 3D cubes environment (with all objects allowed to be moved, i.e., none are fixed in place). Edge color denotes action type. The abstract state graph is nearly identical for each object, which illustrates that the model successfully represents objects in the same manner despite their visual differences.

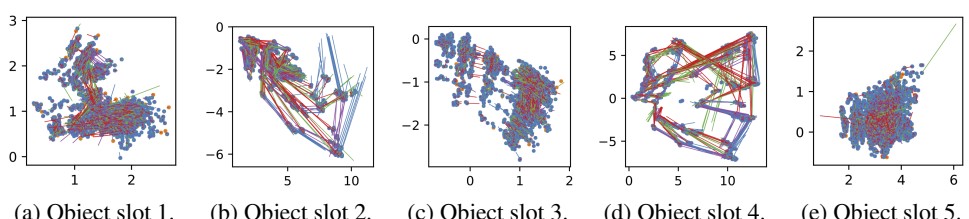

(a) Object slot 1.    (b) Object slot 2.    (c) Object slot 3.    (d) Object slot 4.    (e) Object slot 5.

Figure 6: Abstract state transition graphs per object slot for a trained SWM model *without contrastive loss*, using instead a loss in pixel space, on the 3D cubes environment. Edge color denotes action type.

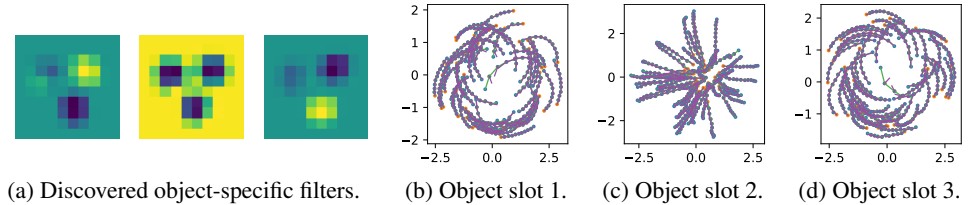

(a) Discovered object-specific filters.    (b) Object slot 1.    (c) Object slot 2.    (d) Object slot 3.

Figure 7: Object filters (left) and abstract state transition graphs per object slot (right) for a trained C-SWM model on unseen test instances of the 3-body physics environment (seed 1).

For the Atari 2600 environments, we generally found latent object representations to be less interpretable. We attribute this to the fact that a) objects have different roles and are in general not exchangeable (in contrast to the block pushing grid world environments and the 3-body physics environment), b) actions affect only one object directly, but many other objects indirectly in two consecutive frames, and c) due to multiple objects in one scene sharing the same visual features. See Figure 9 for an example of learned representations in Atari Pong and Figure 10 for an example in Space Invaders.

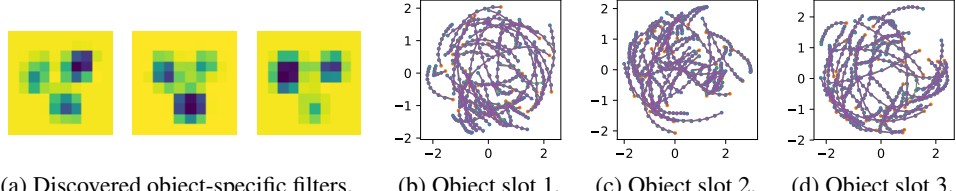

(a) Discovered object-specific filters.  (b) Object slot 1.  (c) Object slot 2.  (d) Object slot 3.

Figure 8: Object filters (left) and abstract state transition graphs per object slot (right) for a trained C-SWM model on unseen test instances of the 3-body physics environment (seed 2).

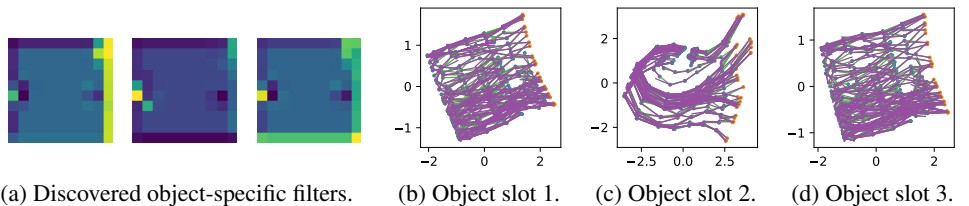

(a) Discovered object-specific filters.  (b) Object slot 1.  (c) Object slot 2.  (d) Object slot 3.

Figure 9: Object filters (left) and abstract state transition graphs per object slot (right) for a trained C-SWM model with $K = 3$ object slots on unseen test instances of the Atari Pong environment.

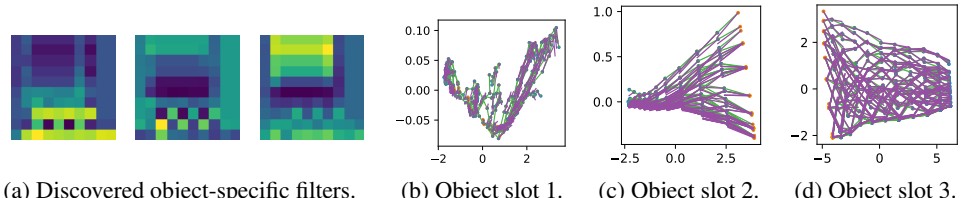

(a) Discovered object-specific filters.  (b) Object slot 1.  (c) Object slot 2.  (d) Object slot 3.

Figure 10: Object filters (left) and abstract state transition graphs per object slot (right) for a trained C-SWM model with $K = 3$ object slots on unseen test instances of the Space Invaders environment.

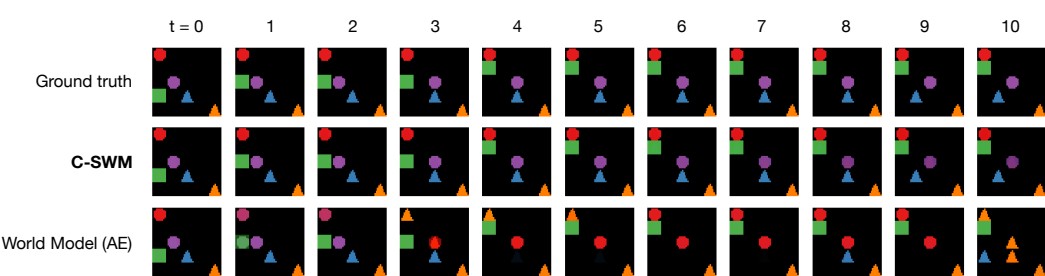

Figure 11: Qualitative model comparison in pixel space on a hold-out test instance of the 2D shapes environment. We train a separate decoder model for 100 epochs on both the C-SWM and the World Model baseline using all training environment instances to obtain pixel-based reconstructions for multiple prediction steps into the future.

## A.2  Model Comparison in Pixel Space

To supplement our model comparison in latent space using ranking metrics, we here show a direct comparison in pixel space at the example of the 2D shapes environment. This requires training a separate decoder model for C-SWM. For fairness of this comparison, we use the same protocol to train a separate decoder for the World Model (AE) baseline (discarding the one obtained from the original end-to-end auto-encoding training procedure). This decoder has the same architecture as in the other baseline models and is trained for 100 epochs.

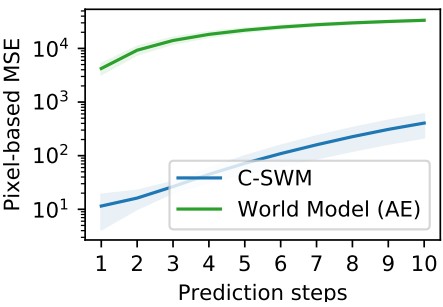

For a qualitative comparison, see Figure 11. The C-SWM model, as expected from our ranking analysis in latent space, performs almost perfectly at this task. Although the World Model (AE) baseline makes clear mistakes which compound over time, it nonetheless often gets several object positions correct after many time steps. The ranking loss in latent space captures this behaviour well, and, for example, assigns an almost perfect score for 1-step prediction to the World Model (AE) baseline. The typically used mean-squared error (MSE) in pixel space (see Figure 12), however, differs by *several orders of magnitude* between the two models and does not capture any of the nuanced differences in qualitative predictive behavior. We hence strongly encourage researchers in this area to consider ranking-based evaluations directly in latent space in addition to, or as a replacement for comparisons in pixel space.

Figure 12: Quantitative model comparison in pixel space on a hold-out test set of the 2D shapes environment. The plot shows mean squared reconstruction error (MSE) in pixel space for multiple transition model prediction steps into the future (lower is better), averaged over 4 runs. Shaded area denotes standard error.

## A.3  Hinge Loss

In Table 2, we summarize results of a comparison between our multi-object contrastive loss from Eq. 6, denoted by *C-SWM (original)*, and a *full-hinge* (or *triplet*) loss that places the hinge over both the positive and the negative energy term, i.e., $\mathcal{L} = \max(0, \gamma + H - \widetilde{H})$. The latter loss is similar to the loss employed in Bordes et al. (2013), but without imposing a norm constraint on the embeddings $z_t$. Reported results are mean and standard error over 4 runs on hold-out environment instances.

Table 2: Comparison of hinge loss variants on 2D shapes environment. Numbers denote ranking scores (in %) for multi-step prediction in latent space. Highest (mean) scores in **bold**.

|  | 1 Step | | 5 Steps | | 10 Steps | |
|---|---|---|---|---|---|---|
| Model | H@1 | MRR | H@1 | MRR | H@1 | MRR |
| **C-SWM (original)** | **100**$_{\pm0.0}$ | **100**$_{\pm0.0}$ | **100**$_{\pm0.0}$ | **100**$_{\pm0.0}$ | **99.9**$_{\pm0.0}$ | **100**$_{\pm0.0}$ |
| with full hinge ($\gamma = 1$) | 95.9$_{\pm1.0}$ | 97.8$_{\pm0.6}$ | 40.4$_{\pm5.9}$ | 54.4$_{\pm5.8}$ | 12.6$_{\pm4.0}$ | 21.8$_{\pm5.9}$ |
| with full hinge ($\gamma = 5$) | 97.7$_{\pm0.9}$ | 98.8$_{\pm0.5}$ | 52.0$_{\pm2.5}$ | 65.7$_{\pm2.6}$ | 15.8$_{\pm2.9}$ | 26.8$_{\pm3.5}$ |
| with full hinge ($\gamma = 10$) | 97.0$_{\pm0.4}$ | 98.5$_{\pm0.2}$ | 49.5$_{\pm1.5}$ | 63.3$_{\pm1.4}$ | 18.7$_{\pm1.2}$ | 30.3$_{\pm1.4}$ |

We find that placing the hinge only on the negative energy term significantly improves results, likely because the full hinge loss can be minimized by simply increasing the norms of the embeddings $z_t$, as outlined in Bordes et al. (2013). This could be addressed by introducing additional constraints, such as by forcing the embeddings $z_t$ to lie on the unit-hypersphere, as done in TransE (Bordes et al., 2013), which we do not consider in this work as most of the environment dynamics in our experiments obey a flat Euclidean geometry (e.g., translation on a grid).

## A.4  Multiple Feature Maps per Object Slot

We experimented with assigning multiple feature maps per object slot to investigate whether this can improve performance on the Space Invaders environment (as an example of one of the more complex

environments considered in this work). All other model settings are left unchanged. Results are summarized in Table 3. The original model variant with one feature map per object slot (using a total of $K = 5$ slots) is denoted by *C-SWM (original)*. Reported results are mean and standard error over 4 runs on hold-out environment instances.

Table 3: Comparison of model variants with multiple feature maps per object on the Space Invaders environment. Numbers denote ranking scores (in %) for multi-step prediction in latent space.

| Model | 1 Step | | 5 Steps | | 10 Steps | |
|---|---|---|---|---|---|---|
| | H@1 | MRR | H@1 | MRR | H@1 | MRR |
| **C-SWM (original)** | $48.5_{\pm 7.0}$ | $66.1_{\pm 6.6}$ | $16.8_{\pm 2.7}$ | $35.7_{\pm 3.7}$ | $11.8_{\pm 3.0}$ | $26.0_{\pm 4.1}$ |
| with 2 feature maps | $48.2_{\pm 7.1}$ | $65.7_{\pm 6.8}$ | $12.8_{\pm 3.0}$ | $27.7_{\pm 3.2}$ | $8.2_{\pm 2.9}$ | $21.2_{\pm 3.4}$ |
| with 5 feature maps | $52.5_{\pm 3.4}$ | $71.0_{\pm 2.6}$ | $18.2_{\pm 4.3}$ | $36.3_{\pm 4.6}$ | $11.8_{\pm 2.5}$ | $25.0_{\pm 3.4}$ |
| with 10 feature maps | $50.0_{\pm 0.9}$ | $68.6_{\pm 1.0}$ | $15.8_{\pm 1.4}$ | $32.4_{\pm 2.3}$ | $9.0_{\pm 1.4}$ | $22.7_{\pm 2.3}$ |

We find that there is no clear advantage in using multiple feature maps per object slot for the Space Invaders environment, but this might nonetheless prove useful for environments of more complex nature.

## A.5 GRID-WORLD ENVIRONMENT VARIANTS

To further evaluate the robustness of our approach, we experimented with two variants of the 2D shapes grid-world environment with five objects. In the first variant, termed *no-op action*, we include an additional 'no-op' action for each object that has no effect. This mirrors the setting present in many Atari 2600 benchmark environments that have an explicit no-op action. In the second variant, termed *random object*, we assign a special property to one out of the five objects: a random action is applied to this object at every turn, i.e., the agent only has control over the other four objects and in every environment step, a total of two objects receive an action (one by the agent, which only affects one out of the first four objects, and one random action on the last object). This is to test how the model behaves in the presence of other agents that act randomly. As our model is fully deterministic, it is to be expected that adding a source of randomness in the environment will pose a challenge.

We report results in Table 4. Reported numbers are mean and standard error over 4 runs on hold-out environment instances. We find that adding a no-op action has little effect on our model, but adding a randomly moving object reduces predictive performance. Performance in the latter case could potentially be improved by explicitly modeling stochasticity in the environment.

Table 4: Comparison of C-SWM on variants of the 2D shapes environment. Numbers denote ranking scores (in %) for multi-step prediction in latent space.

| Model | 1 Step | | 5 Steps | | 10 Steps | |
|---|---|---|---|---|---|---|
| | H@1 | MRR | H@1 | MRR | H@1 | MRR |
| **C-SWM (original env.)** | $100_{\pm 0.0}$ | $100_{\pm 0.0}$ | $100_{\pm 0.0}$ | $100_{\pm 0.0}$ | $99.9_{\pm 0.0}$ | $100_{\pm 0.0}$ |
| + no-op action | $100_{\pm 0.0}$ | $100_{\pm 0.0}$ | $100_{\pm 0.0}$ | $100_{\pm 0.0}$ | $99.9_{\pm 0.0}$ | $99.9_{\pm 0.0}$ |
| + random object | $98.4_{\pm 0.1}$ | $99.2_{\pm 0.0}$ | $95.9_{\pm 0.2}$ | $97.6_{\pm 0.1}$ | $90.4_{\pm 0.3}$ | $93.7_{\pm 0.2}$ |

## A.6 STABILITY

The discovered object representations and identifications can vary between runs. While we found that this process is very stable for the simple grid world environments where actions only affect a single object, we found that results can be initialization-dependent on the Atari environments, where actions can have effects across all objects, and the 3-body physics simulation (see Figures 7 and 8), which does not have any actions. In some cases, the discovered representation can be less suitable for forward prediction in time or for generalization to novel scenes, which explains the variance in some of our results on these datasets.

### A.7 TRAINING TIME

We found that the overall training time of the C-SWM model was comparable to that of the World Model baseline. Both C-SWM and the World Model baseline trained for approx. 1 hour wall-clock time on the 2D shapes dataset, approx. 2 hours on the 3D cubes dataset, and approx. 30min on the 3-body physics environment using a single NVIDIA GTX1080Ti GPU. The Atari Pong and Space Invaders models trained for typically less than 20 minutes. A notable exception is the PAIG baseline model (Jaques et al., 2019) which trained for approx. 6 hours on a NVIDIA TitanX Pascal GPU using the recommended settings by the authors of the paper.

## B DATASETS

### B.1 GRID WORLDS

To generate an experience buffer for training, we initialize the environment with random object placements and uniformly sample an object and an object-specific action at every time step.

We provide state observations as $50 \times 50 \times 3$ tensors with RGB color channels, normalized to $[0, 1]$. Actions are provided as a 4-dim one-hot vector (if an action is applied) or a vector of zeros per object slot in the environment. The action one-hot vector encodes the directional movement action applied to a particular object, or is represented as a vector of zeros if no action is applied to a particular object. Note that only a single object receives an action per time step. For the Atari environments, we provide a copy of the one-hot encoded action vector to every object slot, and for the 3-body physics environment, which has no actions, we do not provide an action vector.

**2D Shapes** This environment is a $5 \times 5$ grid world with 5 different objects placed at random positions. Each location can only be occupied by at maximum one object. Each object is represented by a unique shape/color combination, occupying $10 \times 10$ pixels on the overall $50 \times 50$ pixel grid. At each time step, one object can be selected and moved by one position along the four cardinal directions. See Figure 2a for an example. The action has no effect if the target location in a particular direction is occupied by another object or outside of the $5 \times 5$ grid. Thus, a learned transition model needs to take pairwise interactions between object properties (i.e., their locations) into account, as it would otherwise be unable to predict the effect of an action correctly.

**3D Blocks** To investigate to what degree our model is robust to partial occlusion and perspective changes, we implement a simple block pushing environment using Matplotlib (Hunter, 2007) as a rendering tool. The underlying environment dynamics are the same as in the 2D Shapes dataset, and we only change the rendering component to make for a visually more challenging task that involves a different perspective and partial occlusions. See Figure 2b for an example.

### B.2 ATARI 2600 GAMES

**Atari Pong** We make use of the Arcade Learning Environment (Bellemare et al., 2013) to create a small environment based on the Atari 2600 game Pong which is restricted to the first interaction between the ball and the player-controlled paddle, i.e., we discard the first couple of frames from the experience buffer where the opponent behaves completely independent of any player action. Specifically, we discard the first 58 (random) environment interactions. We use the PONGDETERMINISTIC-V4 variant of the environment in OpenAI Gym (Brockman et al., 2016). We use a random policy and populate an experience buffer with 10 environment interactions per episode, i.e., $T = 10$. An observation consists of two consecutive frames, cropped (to exclude the score) and resized to $50 \times 50$ pixels each. See Figure 2c for an example.

**Space Invaders** This environment is based on the Atari 2600 game Space Invaders, using the SPACEINVADERSDETERMINISTIC-V4 variant of the environment in OpenAI Gym (Brockman et al., 2016), and processed / restricted in a similar manner as the Pong environment. We discard the first 50 (random) environment interactions for each episode and only begin populating the experience buffer thereafter. See Figure 2d for an example observation.

### B.3  3-BODY PHYSICS

The 3-body physics simulation environment is an interacting system that evolves according to classical gravitational dynamics. Different from the other environments considered here, there are no actions. This environment is adapted from Jaques et al. (2019) using their publicly available implementation[1], where we set the step size (dt) to 2.0 and the initial maximum $x$ and $y$ velocities to 0.5. We concatenate two consecutive frames of $50 \times 50$ pixels each to provide the model with (implicit) velocity information. See Figure 4 for an example observation sequence.

## C  EVALUATION METRICS

### C.1  HITS AT RANK K (H@K)

This score is 1 for a particular example if the predicted state representation is in the k-nearest neighbor set around the encoded true observation, where we define the neighborhood of a node to include the node itself. Otherwise this score is 0. In other words, this score measures whether the rank of the predicted state representation is smaller than or equal to k, when ranking all reference state representations by distance to the true state representation. We report the average of this score over a particular evaluation dataset.

### C.2  MEAN RECIPROCAL RANK (MRR)

This score is defined as the average inverse rank, i.e., $\mathrm{MRR} = \frac{1}{N} \sum_{n=1}^{N} \frac{1}{\mathrm{rank}_n}$, where $\mathrm{rank}_n$ is the rank of the $n$-th sample.

## D  ARCHITECTURE AND HYPERPARAMETERS

### D.1  OBJECT EXTRACTOR

For the 3D cubes environment, the object extractor is a 4-layer CNN with $3 \times 3$ filters, zero-padding, and 16 feature maps per layer, with the exception of the last layer, which has $K = 5$ feature maps, i.e., one per object slot. After each layer, we apply BatchNorm (Ioffe & Szegedy, 2015) and a $\mathrm{ReLU}(x) = \max(0, x)$ activation function. For the 2D shapes environment, we choose a simpler CNN architecture with only a single convolutional layer with $10 \times 10$ filters and a stride of 10, followed by BatchNorm and a ReLU activation or LeakyReLU (Xu et al., 2015) for the Atari 2600 and physics environments. This layer has 16 feature maps and is followed by a channel-wise linear transformation (i.e., a $1 \times 1$ convolution), with 5 feature maps as output. For both models, we choose a sigmoid activation function after the last layer to obtain object masks with values in $(0, 1)$. We use the same two-layer architecture for the Atari 2600 environments and the 3-body physics environment, but with $9 \times 9$ filters (and zero-padding) in the first layer, and $5 \times 5$ filters with a stride of 5 in the second layer.

### D.2  OBJECT ENCODER

After reshaping/flattening the output of the object extractor, we obtain a vector representation per object (2500-dim for the 3D cubes environment, 25-dim for the 2D shapes environment, and 1000-dim for Atari 2600 and physics environments). The object encoder is an MLP with two hidden layers of 512 units and each, followed by ReLU activation functions. We further use LayerNorm (Ba et al., 2016) before the activation function of the second hidden layer. The output of the final output layer is 2-dimensional (4-dimensional for Atari 2600 and physics environments), reflecting the ground truth object state, i.e., the object coordinates in 2D (although this is not provided to the model), and velocity (if applicable).

---

[1] https://github.com/seuqaj114/paig

### D.3 TRANSITION MODEL

Both the node and the edge model in the GNN-based transition model are MLPs with the same architecture / number of hidden units as the object encoder model, i.e., two hidden layers of 512 units each, LayerNorm and ReLU activations.

### D.4 LOSS FUNCTION

The margin in the hinge loss is chosen as $\gamma = 1$. We further multiply the squared Euclidean distance $d(x, y)$ in the loss function with a factor of $0.5/\sigma^2$ with $\sigma = 0.5$ to control the spread of the embeddings. We use the same setting in all experiments.

### D.5 BASELINES

**World Model Baseline** The World Model baseline is trained in two stages: First, we train an auto-encoder or a VAE with a 32-dim latent space, where the encoder is a CNN with the same architecture as the object extractor used in the C-SWM model, followed by an MLP with the same architecture as the object encoder module in C-SWM on the flattened representation of the output of the encoder CNN. The decoder exactly mirrors this architecture where we replace convolutional layers with deconvolutional layers. We verified that this architecture can successfully build representations of single frames.

Example reconstructions from the latent code are shown in Figure 13. We experimented both with mean squared error and binary cross entropy (using the continuous channel values in [0, 1] as targets) as reconstruction loss in the (V)AE models, both of which are typical choices in most practical implementations. We generally found binary cross entropy to be more stable to optimize and to produce better results, which is why we opted for this loss in all the baselines using decoders considered in the experimental section.

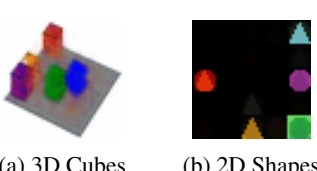

(a) 3D Cubes    (b) 2D Shapes

Figure 13: Reconstructions from the latent code of a trained VAE-based World Model baseline.

In the second stage, we freeze the model parameters of the auto-encoder and train a transition model with mean-squared error on the latent representations. For the VAE model, we use the predicted mean values of the latent representations. This transition model takes the form of an MLP with the same architecture and number of hidden units as the node model in C-SWM. We experimented both with a translational transition model (i.e., the transition model only predicts the latent state difference, instead of the full next state) and direct prediction of the next state. We generally found that the translational transition model performed better and used it throughout all the reported results.

The World Model baselines are trained with a smaller batch size of 512, which slightly improved performance and simplified memory management.

**Ablations** We perform the following ablations: 1) we replace the latent GNN with an MLP (per object, i.e., we remove the edge update function) to investigate whether a structured transition model is necessary, 2) we remove the state factorization and embed the full scene into a single latent variable of higher dimensionality (original dimensionality $\times$ number of original object slots) with an MLP as transition model, and 3) we replace the contrastive loss with a pixel-based reconstruction loss on both the current state and the predicted next state (we add a decoder that mirrors the architecture of the encoder).

**Physics-as-Inverse-Graphics (PAIG)** For this baseline, we train the PAIG model from Jaques et al. (2019) with the code provided by the authors[2] on our dataset with the standard settings recommended by the authors for this particular task, namely: model=PhysicsNet, epochs=500, batch_size=100, base_lr=1e-3, autoencoder_loss=5.0, anneal_lr=true, color=true, and cell_type=gravity_ode_cell. We use the same input size as in C-SWM, i.e., frames are of shape

---

[2]https://github.com/seuqaj114/paig

$50 \times 50 \times 3$. We further set input_steps=2, pred_steps=10 and extrap_steps=0 to match our setting of predicting for a total of 10 frames while conditioning on a pair of 2 initial frames to obtain initial position and velocity information. We train this model with four different random seeds. For evaluation, we extract learned position representations for all frames of the test set and further run the latent physics prediction module (conditioned on the first two initial frames) to obtain model predictions. We further augment position representations with velocity information by taking the difference between two consecutive position representations and concatenating this representation with the 2-dim position representation, which we found to slightly improve results. In total, we obtain a 12-dim (4-dim $\times$ 3 objects) representation for each time step, i.e., the same as C-SWM. From this, we obtain ranking metrics. We found that one of the training runs (seed=2) collapsed all latent representations to a single point. We exclude this run in the results reported in Table 1, i.e., we report average and standard error over the three other runs only.

