# OpenReview forum: "Contrastive Learning of Structured World Models"
_ICLR.cc/2020/Conference — Accept (Talk)_

### Official Review · AnonReviewer3 · 2019-10-06
**Official Blind Review #3**

**Rating:** 8

**Review:**

The construction and learning of structured world models is an interesting area of research that could in principle enable better generalisation and interpretability for predictive models. The authors overcome the problem of using pixel-based losses (a common issue being reconstruction of small but potentially important objects) by using a contrastive latent space. The model otherwise makes use of a fixed number of object slots and a GNN transition model, similarly to prior approaches. The authors back up their method with nice results on 3D cubes and 3-body physics domains, and reasonable initial results on two Atari games, with ablations on the different components showing their contributions, so I would give this paper an accept.

The comparisons to existing literature and related areas is very extensive, with interesting pointers to potential future work - particularly on the transition model and graph embeddings. As expected, the object-factorized action space appears to work well for generalisation, and could be extended/adapted, but setting a fixed number of objects K is a clearly fundamentally limiting hyperparameter, and so showing how the model performs under misspecification of this hyperparameter is useful to know for settings where this is known (2D shapes, 3D blocks, 3-body physics). The fact that K=1 is the best for Pong but K=5 is the best for Space Invaders raises at least two questions: can scaling K > 5 further improve performance on Space Invaders, and is it possible to make the model more robust to a greater-than-needed number of object slots? On a similar note, the data collection procedure for the Atari games seems to indicate that the model is quite sensitive to domains where actions rarely have an impact on the transition dynamics, or the interaction is more complex (e.g. other agents exist in the world) - coming up with a synthetic dataset where the importance of this can be quantified would again aid understanding of the authors' proposed method.

**Experience Assessment:**

I have read many papers in this area.

**Review Assessment: Checking Correctness Of Derivations And Theory:**

N/A

**Review Assessment: Checking Correctness Of Experiments:**

I carefully checked the experiments.

**Review Assessment: Thoroughness In Paper Reading:**

I read the paper thoroughly.

---

> ### Author Response · Authors · 2019-11-11
> **Response to review #3**
>
> Thank you for your valuable feedback.
>
> Please find our responses to your questions and comments below.
>
> [Number of object slots]:
> We have carried out an analysis for the 3-body physics environment with misspecification of the number of object slots (all other parameters are left the same), see results table for mean MRR (in %) results over 4 model runs below. The results show that K=1 is not sufficient for good generalization on the 3-body system, whereas K=5 performs comparable or slightly better than K=3 (note that there is some variance in between runs). Looking only at the best 10-step prediction result out of 4 runs, we have 95.4 (for K=3) vs. 93.3 (for K=5).
>
> +----------------------------------------------------------------+
> | Model                       | 1 Step | 5 Steps | 10 Steps |
> +----------------------------------------------------------------+
> | K=1                           |  97.5    |  71.5    |  40.9     |
> +----------------------------------------------------------------+
> | K=3 (default)             | 100      |  98.5    |  85.2     |
> +----------------------------------------------------------------+
> | K=5                           | 100      |  98.7     |  91.0     |
> +----------------------------------------------------------------+
>
> In terms of increasing K beyond 5 on the Space Invaders benchmark: We have carried out an additional experiment with K=7 with the following mean MRR (in %) results: 71.5 (1 step), 28.3 (5 steps), 22.7 (10 steps) -- i.e., worse in long-term prediction than for K=5 (best setting) but comparable to K=5 in the 1 step prediction setting.
>
> Note that for the block pushing environments, we cannot change the number of object slots without changing the way actions are factorized and presented to our model. To make the model more robust to the number of object slots, an iterative object detection mechanism such as in MONet (Burgess et al., 2019) might be useful which can assign 'empty' slots if all objects are explained by the model already, which would be interesting to explore in future work.
>
> [Synthetic dataset to test the effect of no-op actions & other agents]:
> This is a great suggestion. We have performed additional experiments on a variant of the block pushing (2D Shapes) environment where a) some actions have no effect (no-op action), and b) one block moves randomly and independent of agent actions. We find that adding a no-op action has little to no effect on the ability of the model to discover objects, learn the underlying grid structure and to generalize to new environment instances. In the other setting (one out of the five objects moves randomly), the model correctly discovers representations for the other 4 objects, but learns a "blank" feature map for the randomly moving object -- prediction performance on the test set is negatively affected by the randomly moving object: 93.7 mean MRR (in %) instead of 100 for 10-step prediction. It would be interesting to extend the C-SWM model to explicitly handle uncertainty in environments in future work to address this gap in performance.
>
> We have posted an updated version of our manuscript. You can find a short summary of our changes in our top-level comment.

---

> > ### Comment · AnonReviewer3 · 2019-11-12
> > **Response**
> >
> > Thank you for the additional details and experiments, as I believe these are useful bits of knowledge for probing how your model performs. It appears that K is not *overly* sensitive to misspecification, and indeed automating this would would be a great avenue for future research. The modified 2D shapes environment for testing no-ops and stochasticity sounds like a great realisation of my suggestion, and, again, the results are very interesting to know!

---

### Official Review · AnonReviewer2 · 2019-10-23
**Official Blind Review #2**

**Rating:** 8

**Review:**

This paper tackles the problem of learning an encoder and transition model of an environment, such that the representation learnt uses an object-centric representation which could favor compositionality and generalisation. This is trained using a contrastive max-margin loss, instead of a generative loss as previously explored. They do not consider RL or follow-up tasks leveraging these representations and transition models yet.
They perform an extensive assessment of their model, with many ablations, on 2 gridworld environments, one physical domain, and on Atari.

The paper is very well motivated, easy to follow, and most of its assumptions and decisions are sensible and well supported. They also provide interesting assessments and insights into the evaluation scheme of such transition models, which would be of interest to many practitioners of this field.

Apart from some issues presented below, I feel that this work is of good quality and would recommend it for acceptance.

1.	The model is introduced in a very clear way, and most decisions seem particularly fair. I found the presentation of the contrastive loss with margin to be clear, and the GraphNet is also well supported (although see question below).  However, two choices are surprising to me and would deserve some clarification and more space in the main text, instead of the Appendix:
	a.	Why does the object extractor only output a scalar mask? This was not extremely clear from reading the main text (and confused me when I first saw Figure 1 and 3a), but as explained in the Appendix, the CNN is forced to output a sigmoid logit between [0, 1] per object channel.
	This seems overly constraining to me, as this restricts the network to only output “1 bit” of information per “object”.
	However, maybe being able to represent other factors of these objects might be necessary to make better predictions?
	This also requires the user to select the number of output channels precisely, or the model might fail. This is visible in the Atari results, where the “objectness” is much less clear.
	 Did you try allowing the encoder to output more features per objects?
	Obviously this would be more complicated and would place you closer to a setting similar to MONet (Burgess et al. 2019) or IODINE (Greff et al. 2019), but this might help a lot.
	b.	It was hard to find the dimensionality D of the abstract representation $z_t$. It is only reported in the Appendix, and is set to $D=2$ for the 2D gridworld tasks and $D=4$ for Atari and the physics environments.  These are quite small, and the fact that they exactly coincide with your assumed sufficient statistics is a bit unfortunate.
	What happens if D is larger? Could you find the optimal D by some means?
2.	The GraphNet makes sense to me, but I wondered why you did not provide $a_t^j$ to $e_t^{(i, j)}$ as well? I could imagine situations where one would need the action to know if an interaction between two slots is required.
3.	Similarly, the fact that the action was directly partitioned per object (except in Atari where it was replicated), seemed slightly odd. Would it still work if it was not directly pre-aligned for the network? I.e. provide $a_t$ as conditioning for the global() module of the GraphNet, and let the network learn which nodes/edges it actually affects.
4.	In your multi-object contrastive loss, how is the mapping between slot k in $z_t$ and $\tilde{z}_t$ performed? Do you assume that a given object (say the red cube) is placed in the same $k$ slot across different scenes/timesteps? This may actually be harder to enforce by the network than expected (e.g. with MONet, there is no such “slot stability”, see [1] for a discussion).
5.	It was unclear to me if the “grid” shown in Figure 3 (b) and 5 is “real”? I.e. are you exactly plotting your $z_t$ embeddings, and they happen to lie precisely along this grid? If yes, I feel this is a slightly stronger result as you currently present, given this means that the latent space has mirrored the transition dynamics in a rather impressive fashion.
6.	Related to that point, I found Figure 3 b) to be slightly too hard to understand and parse. The mapping of the colours of the arrows is not provided, and the correspondence between “what 3D object is actually moving where” and “which of the coloured circles correspond to which other cubes in the image” is hard to do (especially given the arbitrary rotation).  Could you add arrows/annotations to make this clearer?  Alternatively, presenting this as a sequence might help: e.g. show the sequence of real 3D images, along with the trajectory it traces on the 2D grid.
7.	Figure 4 a) was also hard to interpret. Seeing these learnt filters did not tell much, and I felt that you were trying too hard to impose meaning on these, or at least it wasn’t clear to me what to take of them directly. I would have left this in the Appendix. Figure 4 b) on the other hand was great, and I would put more emphasis on it.
8.	There are no details on how the actual test data used to generate Table 1 was created, and what “unseen environment instances” would correspond to. It would be good to add this to the Appendix, and point forward to it at the end of the first paragraph of Section 4.6, as if you are claiming that combinatorial generalization is being tested this should be made explicit. I found Table 1 to be great, complete, and easy to parse.
9.	It would be quite interesting to discuss how your work relates to [1], as the principles and goals are quite similar.  On a similar note, if you wanted to extend your 2D shape environment from a gridworld to a continuous one with more factors of variations, their Spriteworld environment [2] might be a good candidate.


References:
[1] Nicholas Watters, Loic Matthey, Matko Bosnjak, Christopher P. Burgess, Alexander Lerchner, “COBRA: Data-Efficient Model-Based RL through Unsupervised Object Discovery and Curiosity-Driven Exploration”, 2019, https://arxiv.org/abs/1905.09275
[2] Nicholas Watters, Loic Matthey, Sebastian Borgeaud, Rishabh Kabra, Alexander Lerchner, “Spriteworld: A Flexible, Configurable Reinforcement Learning Environment”, https://github.com/deepmind/spriteworld/



**Experience Assessment:**

I have published in this field for several years.

**Review Assessment: Checking Correctness Of Derivations And Theory:**

I carefully checked the derivations and theory.

**Review Assessment: Checking Correctness Of Experiments:**

I carefully checked the experiments.

**Review Assessment: Thoroughness In Paper Reading:**

I read the paper thoroughly.

---

> ### Author Response · Authors · 2019-11-11
> **Response to review #2**
>
> Thank you for your extensive review and for your detailed feedback, this is greatly appreciated.
>
> Please find our responses to your questions and comments below.
>
> Q1a [Scalar mask]:
> This is indeed a very good point, which we initially did not try experimentally to avoid additional complexity. Our synthetic block pushing environment does not require more than a simple scalar mask, as the model only needs to encode object location. For more complex environments, it could indeed be beneficial to assign more than one output channel per object. Note that the object encoder (which maps from scalar mask to object latent variable) is an MLP with high-dimensional hidden representations, which allows the model to extract, e.g., object position from its mask. We carried out additional experiments on the Space Invaders task with {2, 5, 10} output channels per object slot and we found no significant difference in MRR results compared to using just one output channel. We added these results to the appendix, and we also further clarified this architecture detail in the paper.
>
> Q1b [Dimensionality of latent space]:
> We ran additional experiments with D>2 on the block pushing experiments (2D shapes) and we found that we get the same results for D in {4, 8, 16}, i.e., MRR=100% (on {1,5,10} prediction steps into the future) and the latent representations lie on a close to perfect 2D grid that is randomly oriented in the higher-dimensional latent space. So it seems like the choice of D does not have a significant influence on results as long as it is not too small. We have improved clarity on this detail in the paper.
>
> Q2 [Actions on edge messages]:
> Thanks for this suggestion. We initially designed the model with the example of the synthetic block pushing environments in mind, where it is not necessary to condition the messages on the action, but this could indeed in principle be useful for the Atari game setting. Alternatively, one could perform multiple rounds of message passing as suggested in the paper. We ran an additional experiment on Space Invaders with K=5 object slots, where we also conditioned the edge update on the action, but the results were very similar to our original setting, where we only condition the node update on the action: 26.0±4.1 MRR (in %) 10-step prediction for the original setting vs. 27.5±2.3 MRR (in %) for the setting with actions included in the edge update.
>
> Q3 [Learning action-to-node assignment]:
> This is a very good point and something we haven't had the chance to try experimentally yet. Extending the GNN model with a global state in the line of GraphNets (Battaglia et al., 2018) would certainly be a good starting point for learning the action-to-node/-edge assignment automatically, but it would likely require some more changes to the model (or to the way actions are encoded) as object slots are fully exchangeable in the current architecture and one would need a way to break this symmetry.
>
> Q4 [Slot stability]:
> Our model is "slot stable" as objects are identified with a particular feature map of the CNN. In other words, we can assume that the same object always ends up in the same slot (for a fixed set of model parameters). While this is convenient, this is of course a limitation as it does not allow for instance disambiguation (e.g. two objects with the same appearance), which needs to be overcome in future work (see "Instance Disambiguation" in Section 4.7 on Limitations). For encoders that are not "slot stable", one could potentially use something like the Sinkhorn distance to compute the energy terms, but this could come with other challenges.
>
> Q5 [Grid-structure of embedding space]:
> Yes, the grid is "real"! There is no post-processing done to get these plots -- we directly visualize the learned 2D embedding space and plot learned transitions as lines/edges. We found it indeed remarkable that the model learns to uncover this latent structure so precisely. We made this point clearer in the paper.
>
> Q6/7 [Figure clarity]:
> Thank you, this is very helpful feedback regarding Figures 3 and 4. We have updated both figures in the paper to improve clarity.
>
> (continued in the next comment due to character limitations)

---

> > ### Author Response · Authors · 2019-11-11
> > **Response to review #2 -- part 2**
> >
> > Q8 [Test set generation]:
> > We generate the test sets in the same way as the training sets (i.e., using a random policy and a random initialization of the environment where possible), but using different random seeds. The state space for the grid world environments is large (~6.4M possible environment configurations), and hence the train/test overlap can be expected to be small (both train and test set contain 100k experience triples, and each test episode is a sequence of 10 such triples). Only for the Atari games the situation is a bit trickier (as episodes can be very similar). We address this by first running a fixed number of random actions before we start populating the experience buffers. We have verified that not a single (full-length) test episode exactly coincides with a training episode, so that doing well on this task requires generalization. We have clarified this in the paper.
> >
> > Q9 [COBRA/Spriteworld]:
> > Thanks for the literature suggestion. The COBRA (Watters et al., 2019) paper is something we have overlooked to include in our related work discussion, and it is indeed very related. The main differences between the unsupervised component of COBRA and our work are that we use a GNN transition model to model interactions (instead of modeling each slot independently with an MLP), a contrastive loss (instead of decoding back into pixel space), and a simpler object detection module (COBRA uses MONet, which would be interesting to try in future work in a contrastive setting as well, but this would require solving a matching problem as outlined in the appendix of the COBRA paper). Lastly, our model is trained end-to-end whereas COBRA is demonstrated using a pre-trained vision model. We have included a short discussion in the paper.
> >
> > The Spriteworld tasks considered in the COBRA paper will most likely pose some challenges to our simplified object detector/encoder, as we do not perform instance disambiguation (but rather assume that there is a fixed number of objects of specific appearance), but we agree that it would be a very interesting benchmark for testing an extension of C-SWM with a more elaborate encoder.
> >
> > We have posted an updated version of our manuscript. You can find a short summary of our changes in our top-level comment.

---

> > > ### Comment · AnonReviewer2 · 2019-11-14
> > > **Response to authors**
> > >
> > > Thank you very much for the extremely thorough rebuttal, and for the extra experiments on such short notice. The improved figures and text are great.
> > >
> > > This improved the manuscript a lot for me and I would recommend it for acceptance.
> > >
> > > Q1a+Q4: Great to know, thanks for checking. The text is also clearer now, along with the discussion of when this would help or not.
> > > Q1b: Very interesting to see that this still works for D>2, thanks a lot for running these.
> > >
> > > --
> > >
> > > One extra question I was interested to hear your thoughts on:
> > >
> > > 10. Why is PAIG doing so poorly on the 3-body problem? Given it has the "true" dynamics model built-in, I would have expected it to perform better?
> > > Their Figure 3 seems to indicate near-perfect results at 10-steps, however it drops to a third of top-performance in your case?

---

> > > > ### Author Response · Authors · 2019-11-15
> > > > **Response to reviewer #2**
> > > >
> > > > Thank you for your kind response and for taking our rebuttal into account for your assessment of our paper.
> > > >
> > > > Regarding your extra question:
> > > >
> > > > Q10 [PAIG baseline]:
> > > > Our dataset slightly differs from that used in PAIG: we use larger time steps between rendered frames in the physical simulator (as we are only using a total of 10 observations, every observation consists of two consecutive frames) and our frames are of slightly different size (50x50x3 as opposed to PAIG’s 36x36x3) to be in line with the other environments we tested. Lastly, we use only 2 frames for velocity estimation instead of 4 for fair comparison with the World Model baseline and C-SWM (which take pairs of 2 frames at every time step).
> > > >
> > > > None of these changes, however, should in principle prevent the PAIG model from learning the correct dynamics. We will have a closer look at what causes the PAIG model to fail (due to very little remaining time we will have to do this after the rebuttal period), but looking at the PAIG model predictions, it looks like the model often fails at object identification (predicting the correct mask for an object) in our setting. You can find a video of PAIG model predictions in our anonymous repository (paig_predictions.gif).

---

> > > > > ### Public Comment · ~Miguel_Jaques1 · 2020-01-08
> > > > > **Responding to the PAIG model comparison (PAIG author here)**
> > > > >
> > > > > Now that both this paper and PAIG have been accepted to ICLR, I can confirm the authors' reason for the failure of PAIG here.
> > > > >
> > > > > Though in principle the changes mentioned should not prevent the PAIG model from learning the correct dynamics, in our experiments we found that for the 3-ball dataset giving 2 frames as input to the model was not sufficient signal for the model to train correctly. This includes discovering the objects (hence the inability to find correct object masks, as the authors mention) and learning the correct physical parameters of the scene.
> > > > >
> > > > > While we agree that in this setting a fair comparison involves only passing 2 frames as input, that is not enough to get a fair comparison with our model in terms of its "full" ability in a non-handicapped setting.

---

### Official Review · AnonReviewer1 · 2019-10-25
**Official Blind Review #1**

**Rating:** 8

**Review:**

This paper aims to learn a structured latent space for images, which is made up of objects and their relations. The method works by (1) extracting object masks via a CNN, (2) turning those masks into feature vectors via an MLP, (3) estimating an action-conditioned delta for each feature via a GNN. Learning happens with contrastive losses, which ask that each feature+delta is close to the true next feature, and far away from other random possibilities. Experiments in simple synthetic environments (e.g., 2D geometric shapes moving on a black background) show encouraging results.

This paper has a simple, well-motivated method. It is clearly written, and easy to understand. The evaluation is straightforward also: the paper merely shows that this model's nearest neighbors in featurespace are better than the nearest neighbors of World Model (2018) and PAIG (2019). Also, some visualizations indicate that for these simple directional manipulations (up/down/left/right motion), PCA compressions of the model's states have a clean lattice-like structure.

It is impressive that the model discovers and segments objects so accurately. Perhaps this could actually be evaluated. However, I do not understand why results are so sensitive to the number of object slots (K). This seems like a severe limitation of the model, since in general we have no idea what value to set for this.

Although I like the paper, I am not sure that there is sufficient evidence for the method being something useful. Yes, H@1 and MRR are high, but as the paper itself implies, the real goal is to improve performance (or, e.g., sample efficiency) in some downstream task. Given how simple these domains are, and the fact that data is collected with purely random exploration, it is difficult to imagine that there is any significant difference between the training set and the test set. For example, if you make 1000 episodes of 10 steps each in Space Invaders, you practically get 1000 copies of the same 10 frames. I worry that all the evaluation has shown so far is that this model can efficiently represent the state transitions that it has observed.

The authors note that it was beneficial to only use the hinge on the negative energy term. This seems unusual, since a hinge on the positive term allows some slack, which intuitively makes the objective better-formulated. Can the authors please clarify this result, at least empirically?


**Experience Assessment:**

I have published one or two papers in this area.

**Review Assessment: Checking Correctness Of Derivations And Theory:**

I assessed the sensibility of the derivations and theory.

**Review Assessment: Checking Correctness Of Experiments:**

I assessed the sensibility of the experiments.

**Review Assessment: Thoroughness In Paper Reading:**

I read the paper thoroughly.

---

> ### Author Response · Authors · 2019-11-11
> **Response to review #1**
>
> Thank you for your valuable feedback.
>
> Please find our responses to your questions and comments below.
>
> [Number of object slots (K)]:
> This is a very good question. Our results indicate that it is best to choose K based on validation set performance if there is no clear a-priori choice. Generalization to unseen environment instances likely not only depends on how well objects are discovered and represented, but also to what degree the learned transition model (GNN) on this structured latent space generalizes. Hence, it is difficult to a-priori predict which number of object slots would work well on a particular problem, unless the model has some built-in mechanism to assign "empty" slots, such as the iterative mechanism in MONet (Burgess et al., 2019), which however relies on pixel-based losses. Despite the dependency on K, we still observe stronger generalization performance across a range of settings compared to unstructured baselines using pixel-based losses.
>
> [Difference between training and test sets]:
> This is a very good point and we indeed try to control for this issue. For the Atari benchmarks, we populate the experience buffer only after a certain number of frames (which represent fully deterministic opponent transitions) during which we take random actions. We have verified that no episode (i.e., the full 10-step sequence of states/actions) in the test set exactly coincides with an episode from the training set for both Pong and Space Invaders, and hence performing well on this task requires some form of generalization. In the grid world / block pushing environments, there are around 6.4M possible environment configurations, and hence the train/test overlap can be expected to be small (both train and test set contain 100k experience triples). For the physics simulation the state space is continuous and starting positions are sampled at random. We have made this clearer in the paper.
>
> [Hinge loss]:
> We performed a direct comparison between our loss and the triplet loss from TransE (Bordes et al., 2013), i.e. with the hinge covering both the positive and the negative energy term. The table below summarizes mean MRR (in %) results (from 4 runs with random init.) on the 2D Shapes environment for hinge parameters $\gamma$ in {1,5,10}.
>
> +----------------------------------------------------------------+
> | Model                            | 1 Step | 5 Steps | 10 Steps |
> +----------------------------------------------------------------+
> | C-SWM (default loss, $\gamma=1$) | 100    | 100     | 100      |
> +----------------------------------------------------------------+
> | Full hinge, $\gamma=1$           | 97.8   | 54.4    | 21.8     |
> +----------------------------------------------------------------+
> | Full hinge, $\gamma=5$           | 98.8   | 65.7    | 26.8     |
> +----------------------------------------------------------------+
> | Full hinge, $\gamma=10$          | 98.5   | 63.3    | 30.3     |
> +----------------------------------------------------------------+
>
> This setting performs significantly worse in our case, most likely because we do not force the embeddings to lie on a (hyper-)sphere (i.e., L2 norm = 1). In (Bordes et al., 2013), the authors include this constraint to avoid pathologies in their loss function (trivial minimization by growing the norms of the embeddings), which might be the cause for suboptimal performance in our case. We do not wish to constrain embeddings to a hypersphere in general, however, as this could affect how accurately we can learn certain structures (e.g., a hyperspherical latent space is likely less suitable for learning grid-structured representations and might make it more difficult for the transition model to generalize). Hyperspherical latent spaces could be useful, however, for learning rotational transformations.
>
> We have posted an updated version of our manuscript. You can find a short summary of our changes in our top-level comment.

---

> > ### Comment · AnonReviewer1 · 2019-11-15
> > **Thank you**
> >
> > Your point about L2 normalization is interesting. Do you have a reference for the claim that "hyperspherical latent space is likely less suitable for learning grid-structured representations"?

---

> > > ### Author Response · Authors · 2019-11-15
> > > **Response to reviewer #1**
> > >
> > > Thank you for your follow-up question.
> > >
> > > The following workshop paper is a good reference for an exploration of this issue in the context of representation learning for relational data:
> > >
> > > Melanie Weber & Maximilian Nickel, "Curvature and Representation Learning: Identifying Embedding Spaces for Relational Data", NeurIPS 2018 Workshop on Relational Representation Learning, https://web.math.princeton.edu/~mw25/project/files/nips_FB.pdf

---

### Author Response · Authors · 2019-11-11
**General response**

We would like to thank the reviewers for their valuable feedback.

We have updated our manuscript with the following changes:
- We have added additional experimental results in Appendix A with a) a variant of our loss function that places the 'hinge' over the energy terms of both positive and negative samples (R1), b) multiple feature maps per object slot (R2), and c) variants of our grid world environments with no-op actions and with randomly moving objects (R3).
- We have improved the clarity of our exposition and writing to address questions and comments by all three reviewers (R1-3).
- We discuss additional related work (R2).
- We have improved clarity of Figures 3 and 4 (R2).

These changes have increased the length of the main part of the paper to close to 9 pages. We believe that this is justified for better coverage of related work and for improved clarity.

---

### Public Comment · ~ALEKSANDR_MIHEEV1 · 2019-12-24
**Just some questions for clear understanding**

Your work is very inspiring! I think this is a very actual work, considering that the problems of classification and segmentation are already well solved. I can’t do a review yet, because I don’t understand a few points. Perhaps this is a language barrier or because some accepted concepts are not obvious to me.

1. Do you generate an explicit output from somewhere or does the model itself receive all the data for training? If your loss function compares hidden states, then where do you get the true values for hidden states? They can be taken only by passing through the model.  Does this mean that this model begins with random values ​​and self-organizes over time, that is, not only the network parameters are trained, but also the output itself converges to the desired value?

2. Do you pretrain the CNN Object extractor or everything modules train automatically end to end? This question continues the first question. I understand that perhaps this is the essence of your work. I read the comments and reviews here, but still I don't have a clear understanding.

3. Do I understand correctly that you are comparing your model with models that work without structure, that is VAE that predict the next state in pixel format? But your model predicts the next state in the hidden vector format that still needs to be transforms in pixel image. But you do not have such module in the Figure 1.

4. RL involves action. Yes, for some agents, you need to predict the next state, but in the end you need to choose an action. Does your model do actions or actions and states (game replays) typed in another way?

---

> ### Author Response · Authors · 2020-01-05
> **Re: Just some questions for clear understanding**
>
> Thank you for your questions, Aleksandr.
>
> Re: 1) The training data only consists of an offline experience buffer. The model does not have access to labeled / ground-truth hidden state data. Hence, the model must infer the hidden state from observational data in the form of observation-action-observation triples only.
>
> Re: 2) We do not pre-train the CNN object extractor.
>
> Re: 3) We compare to two baselines, one with (PAIG) and one without (World Model) object-factorized hidden states. One major difference between our model (C-SWM) and the baselines is that we do indeed not use a decoder back into pixel space.
>
> Re: 4) C-SWM learns an action-conditioned transition model in latent space. We do not do reinforcement learning (RL) in this work.

---

### Decision · Program_Chairs · 2019-12-19

**Decision:**

Accept (Talk)

**Comment:**

This paper presents an approach to learn state representations of the scene as well as their action-conditioned transition model, applying contrastive learning on top of a graph neural network. The reviewers unanimously agree that this paper contains a solid research contribution and the authors' response to the reviews further clarified their concerns.